# A Metabolomics Analysis of Postmenopausal Breast Cancer Risk in the Cancer Prevention Study II

**DOI:** 10.3390/metabo11020095

**Published:** 2021-02-10

**Authors:** Steven C. Moore, Kaitlyn M. Mazzilli, Joshua N. Sampson, Charles E. Matthews, Brian D. Carter, Mary C. Playdon, Ying Wang, Victoria L. Stevens

**Affiliations:** 1Division of Cancer Epidemiology and Genetics, National Cancer Institute, Rockville, MD 20895, USA; kmm509@sph.rutgers.edu (K.M.M.); Joshua.Sampson@nih.gov (J.N.S.); Charles.Matthews2@nih.gov (C.E.M.); 2Department of Population Science, American Cancer Society, Atlanta, GA 30303, USA; brian.carter@cancer.org (B.D.C.); ying.wang@cancer.org (Y.W.); vstevens311@gmail.com (V.L.S.); 3Cancer Control and Population Sciences, Huntsman Cancer Institute, Salt Lake City, UT 84112, USA; Mary.Playdon@hci.utah.edu; 4Department of Nutrition and Integrative Physiology, University of Utah, Salt Lake City, UT 84112, USA

**Keywords:** breast cancer, epidemiology, carnitines, glycerolipids, sex steroid hormones, alcohol, branched-chain amino acids

## Abstract

Breast cancer is the most common cancer in women, but its incidence can only be partially explained through established risk factors. Our aim was to use metabolomics to identify novel risk factors for breast cancer and to validate recently reported metabolite-breast cancer findings. We measured levels of 1275 metabolites in prediagnostic serum in a nested case-control study of 782 postmenopausal breast cancer cases and 782 matched controls. Metabolomics analysis was performed by Metabolon Inc using ultra-performance liquid chromatography and a Q-Exactive high resolution/accurate mass spectrometer. Controls were matched by birth date, date of blood draw, and race/ethnicity. Odds ratios (ORs) and 95% confidence intervals (CIs) of breast cancer at the 90th versus 10th percentile (modeled on a continuous basis) of metabolite levels were estimated using conditional logistic regression, with adjustment for age. Twenty-four metabolites were significantly associated with breast cancer risk at a false discovery rate <0.20. For the nine metabolites positively associated with risk, the ORs ranged from 1.75 (95% CI: 1.29–2.36) to 1.45 (95% CI: 1.13–1.85), and for the 15 metabolites inversely associated with risk, ORs ranged from 0.59 (95% CI: 0.43–0.79) to 0.69 (95% CI: 0.55–0.87). These metabolites largely comprised carnitines, glycerolipids, and sex steroid metabolites. Associations for three sex steroid metabolites validated findings from recent studies and the remainder were novel. These findings contribute to growing data on metabolite-breast cancer associations by confirming prior findings and identifying novel leads for future validation efforts.

## 1. Introduction

Breast cancer is the most frequently diagnosed cancer among women, with 2 million cases globally in 2018, and rates rising rapidly [1]. Since incidence rates have historically been two to three times higher in developed than developing nations, it has been thought that “westernization” and accompanying lifestyle changes underlie much of breast cancer risk [2,3]. By the early 2000s, epidemiologists had identified risk factors—such as shorter durations of breastfeeding—that explain up to half the excess risk within developed nations [4,5]. Explanations for the remaining half, however, have proven elusive.

Several lines of evidence indicate that metabolism may be important to breast cancer etiology, including the established associations of physical inactivity and obesity with breast cancer risk [6,7], and the general role that perturbed metabolism plays as a hallmark of cancer [8,9]. Only recently, however, have studies begun to evaluate metabolism more systematically in relation to breast cancer risk, primarily using metabolomics [10,11,12,13,14,15]. These studies reported that levels of many different metabolites were associated with the risk of breast cancer, including metabolites related to mitochondrial and branched-chain amino acid metabolism [14,15], sex steroid hormones [13,14], and various dietary factors [13]. However, some studies were small (e.g., 621 or fewer cases) [10,11,13,14] and therefore in need of replication, while others examined only a small set of metabolites (e.g., <200) [11,12,15] relative to what is currently possible on some metabolomics platforms.

In the present study, we examined levels of 1256 metabolites in prediagnostic serum in a nested case-control study of 782 postmenopausal breast cancer cases and 782 matched controls in the Cancer Prevention Study-II (CPS-II) Nutrition Cohort. The broad span of metabolites examined allowed us to simultaneously evaluate prior metabolic findings for replication and to search for novel metabolic risk factors that may not have been examined in studies preceding the advent of metabolomics.

## 2. Results

Study participants were an average (mean) 68 years of age and 98% were white (Table 1). The established risk factors for breast cancer of high BMI, early age at menarche, late age at menopause, late age at first live birth/parity, a history of benign breast disease, and family history of breast cancer were (as expected) more prevalent in cases than in controls.

Twenty-four of the 1275 metabolites were statistically significantly associated with breast cancer at a false discovery rate <0.20 in conditional logistic regression models (one below the Bonferroni threshold of 0.00004) (Figure 1; complete results in Appendix A). These metabolites were predominantly lipids (n = 14), followed by metabolites of unknown identity (n = 6), cofactors/vitamins (n = 2), amino acids (n = 1), and a xenobiotic (n = 1). No carbohydrates, energy, or peptide metabolites were significantly associated with breast cancer. Nine of the metabolites had a positive association with risk, with odds ratios (ORs) for a 90th versus 10th percentile comparison that ranged from 1.75 (95% confidence interval (CI): 1.29–2.36) for X-24293 to 1.45 (95% CI: 1.13–1.85) for 16alpha-hydroxy DHEA 3-sulfate, and 15 metabolites had an inverse association with risk, with ORs ranging from 0.59 (95% CI: 0.43–0.79) for 3,4-methyleneheptanoylcarnitine to 0.69 (95% CI: 0.55–0.87) for oxalate (Figure 2).

After confirming that results for unconditional models were like those of conditional models (the median absolute difference in OR between models was 0.01), we further examined associations between metabolites and breast cancer among non-hormone-users. We found that five metabolites were associated with breast cancer risk in this group at FDR < 0.2 (one below the Bonferroni threshold). The metabolite X-11795 was associated with breast cancer only in non-hormone-users, while the other four metabolites were associated with risk in both non-hormone-users and in all participants (X-24293, 1-palmitoyl-2-palmitoleoyl-GPC (16:0/16:1), Androstenediol (3beta,17beta) monosulfate (2), and 4-allylphenol sulfate) (Figure 3). The respective breast cancer ORs for these metabolites were 2.58 (95% CI: 1.70–3.90), 2.22 (95% CI: 1.45–3.40), 2.18 (95% CI: 1.40–3.39), 1.93 (95% CI: 1.32–2.82), and 0.45 (95% CI: 0.30–0.67). The heterogeneity by menopausal hormone use (current vs non-current use), was statistically significant for X-11795 (p_heterogeneity_ = 7.34 × 10^−6^; q_heterogeneity_ = 0.07). In our analysis of estrogen receptor-positive (ER+) cases, no associations met the FDR threshold for statistical significance. Results for the 20 metabolites most significantly associated with breast cancer among non-hormone-users and ER+ breast cancer are shown in Appendix A. In exploratory analyses, associations did not vary by time to diagnosis (<5 years vs ≥5 years) or by cancer stage (localized vs regional/distant) (all q-values > 0.2).

Adjusting for additional covariates in our statistical models had little effect on the magnitude of metabolite-breast cancer associations, suggesting that associations are largely independent of known breast cancer risk factors (Table 2). Fifteen of the 24 associations changed by less than 5% when adjusting for these additional factors. The largest change was −9.7% for X-24293 (OR_age-adjusted_ = 1.75 vs OR_multivariable_ = 1.58).

16alpha-hydroxy DHEA 3-sulfate and three other androgen metabolites are biochemically related to the hormone DHEA and, in our study, each of these metabolites was highly correlated with DHEA-sulfate, the body’s major reservoir of DHEA. The Pearson correlation of 16alpha-hydroxy DHEA 3-sulfate with DHEA-sulfate, for example, was 0.71. To test whether the associations of these metabolites with breast cancer risk were independent of DHEA, we added DHEA-sulfate to models but found little attenuation of associations. For example, the OR for 16alpha-hydroxy DHEA 3-sulfate was 1.54 (95% CI: 1.09–2.19) after adding DHEA-Sulfate, when compared with 1.45 (95% CI: 1.13–1.85) previously. In contrast, adding 16alpha-hydroxy DHEA 3-sulfate to models did affect the association of DHEA sulfate with breast cancer risk. The OR for DHEA sulfate was 0.91 (95% CI: 0.64–1.30) when 16alpha-hydroxy DHEA 3-sulfate was in the model, but 1.24 (95% CI: 0.97–1.60) when 16alpha-hydroxy DHEA 3-sulfate was not in the model.

In a metabolite correlation heatmap, we identified four distinct metabolite clusters encompassing 17 of the 24 breast cancer-associated metabolites (Figure 4). The first and largest cluster included four unknown metabolites (X-11478, X-21319, X-16944, X-18921) and three carnitines (3,4-methyleneheptanoylcarnitine, linolenoylcarnitine (C18:3), and linolenoylcarnitine (C18:2)). The second cluster comprised the four sex steroid hormones of androstenediol (3beta,17beta) monosulfate (2), androsteroid monosulfate (1), androstenediol (3beta,17beta) disulfate (1), and 16alpha-hydroxy DHEA 3-sulfate. The third cluster consisted of the three lipids of 1-(1-enyl-oleoyl)-GPE (P-18:1), 1-(1-enyl-stearoyl)-GPE (P-18:0), 1-(1-enyl-palmitoyl)-2-linoleoyl-GPC (P-16:0/18:2). The fourth and final cluster included the lipids of 1-palmitoyl-2-palmitoleoyl-GPC (16:0/16:1), 2-palmitoleoyl-GPC (16:1), 1-palmitoyl-2-oleoyl-GPC (16:0/18:1). Within each cluster, the correlations between metabolites ranged from 0.32 to 0.89.

## 3. Discussion

After controlling for multiple testing, we found that levels of 24 distinct metabolites were associated with the risk of postmenopausal breast cancer. Findings for three of the metabolites—16alpha-hydroxy DHEA 3-sulfate, androstenediol (3beta,17beta) monosulfate (2), and androstenediol (3beta,17beta) disulfate (1)—replicate results from recent studies [13,14], suggesting these associations are reliable and worth further exploring. The other findings consist of novel associations for carnitines, glycerophosphocholines, and glycerophosphoethanolamines that future studies should aim to replicate. Several associations, particularly those for hormonal metabolites, appeared stronger among women not using menopausal hormones at baseline, though the difference was statistically significant only for X-11795. Associations did not appear to vary by ER status. Overall, our study contributes to the growing data on metabolite-breast cancer associations by replicating three prior associations and by providing novel leads for future studies to replicate—findings that may help us to better understand breast cancer’s underlying mechanistic basis.

The three metabolites with replicating associations each share a common feature, namely DHEA as a precursor. While DHEA has previously been linked with breast cancer risk DHEA [16,17,18], the associations we identified were statistically independent of DHEA levels, much like in one of our prior analyses [14]. This suggests that the pathways these metabolites are part of could constitute novel targets for prevention efforts and that knowing levels of these metabolites could at least nominally improve breast cancer risk prediction models. Mechanistically, increased formation of 16alpha-hydroxy DHEA 3-sulfate, relative to other DHEA metabolites, may increase breast cancer risk because it can be further metabolized into 16alpha-hydroxyestrone and another 16-hydroxylation pathway estrogen metabolites [19], which studies suggest are especially carcinogenic [20,21]. The other two metabolites are derived specifically from 4-androstenediol, a DHEA metabolite that is more structurally similar to testosterone than other androstenediols [22]. Possibly, increased formation of 4-androstenediol and its metabolites, relative to other forms of androstenediol, may be particularly related to breast cancer risk. Of the seven 3β-androstenediol metabolites (e.g., 5alpha-androstan-3beta,17alpha-diol disulfate) we examined, none were associated with breast cancer risk, hinting at some specificity of association.

Eighteen other associations were novel to the literature and generally involved metabolites whose biological function is understood incompletely. One cluster included the metabolites of linoleoylcarnitine and linolenoylcarnitine, which are long-chain acyl fatty acid derivative esters of carnitines. Levels of such derivatives in serum can reflect shifts in mitochondrial metabolism of fats and proteins [23,24]. Two other clusters included glycerophosphocholines and glycerophosphoethanolamines, and circulating levels of these may indicate perturbations in the formation and maintenance of cellular membranes [25]. Additionally, cancer metabolism studies implicate perturbed choline metabolism as a potential metabolic hallmark of oncogenesis [26]. The last cluster included the sex steroid hormone metabolites, whose relevance was discussed above. If associations for these metabolites replicate, further targeted analyses will be needed to clarify the mechanistic basis of these relationships. An additional three associations--for threonate, oxalate, and 4-allyphenol sulfate—were examined in previous studies and were not statistically significant [13,14].

In total, five prospective studies on metabolomics and breast cancer preceded our own and they reported that an aggregate of 51 metabolites was associated with risk of breast cancer at the FDR < 0.2 level of statistical significance [10,11,12,13,14,15]. We evaluated 27 of these in relation to breast cancer, however, after control for multiple testing, only the three DHEA metabolites discussed above had replicated associations. Five further metabolites replicated at the nominal level of statistical significance (*p* < 0.05)—namely threonate, a marker for dietary supplement use [13]; two 4-androstenediol metabolites (androstenediol (3beta,17beta) monosulfate-1 and androstenediol (3beta,17beta) disulfate-2); 3-methyl-2-oxobutyrate, a branched-chain amino acid metabolite; and alpha-hydroxyisovalerate, an alcohol-related metabolite [13]. The remaining 19 metabolites had no discernable association with breast cancer risk in these data. If we include replication at the nominal level of significance, our results suggest an approximate 30% replication rate of prior findings, a less-than-ideal rate that indicates that large studies and/or meta-analyses may be needed in this research area [27,28,29].

Strengths of our study include a large number of cases and controls, prospective collection of serum, the detailed data on breast cancer risk factors, and the use of a reliable and robust metabolomics platform. The large number of metabolites we assessed made it possible to both identify novel associations and to attempt replication of previously reported findings.

Our study also has several limitations. We analyzed blood samples from a single point in time for each participant. Since metabolite levels can change over time, this may cause attenuation of odds ratios [28]. We did not use fasting samples, though our prior data suggest that few metabolites are strongly related (>10% of variance explained) to recent food intake [28]. The sample size among non-hormone-users was relatively small at 319 cases. Our study was observational and so unmeasured lifestyle or biological factors could confound associations. Additionally, though we adjusted for parity, we could not directly adjust for months of breastfeeding, which is an important breast cancer risk factor [4]. Nevertheless, our fully adjusted models differed little from age-adjusted models, suggesting that at least traditional breast cancer risk factors are not major confounders here. Our sample consisted predominately of non-Hispanic white women and results may not generalize to other women. Finally, the novel findings from our study require replication in other cohorts or in a consortium of cohorts, like the Consortium of Metabolomics Studies [29].

In summary, our metabolomics analysis delivered a strong replication of associations between DHEA metabolites and risk of breast cancer and identified novel associations that, if replicated, may provide important clues about breast cancer’s mechanistic basis. The study also provided some limited support for the notion that branched-chain amino acids and alcohol-related metabolites are associated with the risk of breast cancer. Future studies are needed to corroborate these findings and to further expand our understanding of the role of metabolism in breast cancer etiology.

## 4. Materials and Methods

### 4.1. Study Population and Design

The CPS-II Nutrition Cohort is a large, well-characterized prospective cohort established in 1992 by the American Cancer Society (ACS) that enrolled 184,185 men and women residing in 21 states with population-based state cancer registries [30]. The CPS-II Nutrition Cohort was a sub-cohort of CPS-II, a prospective study of cancer mortality among roughly 1.2 million adults begun in 1982. From June 1998 through May 2001, 21,963 of these women donated nonfasting blood samples [30]. Samples were aliquoted, and frozen in liquid nitrogen at approximately −130 °C for long-term storage. These women also completed a self-administered questionnaire at blood draw that assessed reproductive and medical history and lifestyle factors.

Our nested case-control analysis includes all 782 of the breast cancer cases (ICD-O C50) that occurred among the postmenopausal women who donated blood from 1998 to 2001. Cases were first self-reported on biennial questionnaires and then verified through medical records (90%) or through state cancer registries (10%) and they occurred a median of five years after blood donation. Stage data were available for 99.6% of cases and 172 cases were at a regional/distant stage. Estrogen receptor status data were available on ~94% of participants and 644 cases were estrogen receptor-positive (ER+). Controls with no history of breast cancer or other cancer (except for nonmelanoma skin cancer) were selected by incidence density sampling and 1:1 matched to cases by birth date (±6 months), race/ethnicity (Caucasian, African American, or other/unknown), and date of blood draw (±6 months). Controls were alive and had no history of cancer as of the date of diagnosis for the matched case. All aspects of the CPS-II Nutrition Cohort were approved by the Emory University (Atlanta, GA, USA) Institutional Review Board.

### 4.2. Metabolomics Assessment

Metabolomics analysis was performed by Metabolon (Metabolon Inc., Durham, NC, USA), whose platform and analytical process have been previously detailed [31,32]. Briefly, serum samples were treated with methanol and two minutes of vigorous shaking followed by centrifugation to precipitate proteins. The resulting extract was split into four fractions that were analyzed using ultra-performance liquid chromatography and a Q-Exactive high resolution/accurate mass spectrometer interfaced with a heated electrospray ionization source and an Orbitrap mass analyzer operated at 35,000 mass resolution. The fractions were examined under the following conditions: (1) acidic positive ion conditions chromatographically optimized for more hydrophilic compounds, (2) acidic positive ion conditions chromatographically optimized for more hydrophobic compounds, (3) basic negative ion optimized conditions, and (4) negative ionization following elution from a hydrophilic interaction liquid chromatography column. Metabolites were identified through comparison with Metabolon’s library of purified standards or recurrent unknown compounds based on retention time/index, mass to charge ratio, and chromatographic data.

Metabolon quantified the relative concentrations of 1385 metabolites in their analysis. Of these, 110 metabolites (primarily drug metabolites) had no detectable level in ≥90% of samples and we excluded them from analyses. Among the remaining 1275 metabolites, 61% of metabolites had fewer than 5% of their values below the limit of detection. Any values below the limit of detection were assigned the lowest value observed for that metabolite in subsequent analyses. To minimize batch variability and account for non-normal distributions, each metabolite’s value was divided by its daily median then log-transformed for statistical analyses.

Triplicates of 46 samples and duplicates of five samples were used as quality controls to assess the reproducibility of the platform. Across the 1275 metabolites, the median intraclass correlation coefficient (ICC) was 0.93 (25th and 75th percentile, respectively: 0.83, 0.98), indicating a high level of technical reliability comparable to that observed in previous analyses [13,28]. The median within-batch coefficient of variation was 0.13 and the median between-batch coefficient of variation was 0.18.

### 4.3. Statistical Analysis

For each metabolite, we estimated the age-adjusted odds ratio (OR) and 95% confidence intervals (CIs) of breast cancer risk using conditional logistic regression to account for matching. Odds ratios for metabolites were scaled so that they represent risk at the 90th percentile of metabolite values when compared with risk at the 10th percentile (OR = *e*^β(X90-X10)^ where β is the coefficient for the metabolite modeled continuously and X_90_ and X_10_ are metabolite values at the 90th and 10th percentiles). Associations were also separately estimated for cases among non-hormone-users and for ER+ cases, as prior studies suggest that metabolite associations may be evident predominantly in non-hormone users and/or for the ER+ cancer subtype [14,15]. For the analysis of non-hormone-users, it was necessary to break matched sets and analyze associations using unconditional logistic regression since cases and controls were not matched on hormone use. The threshold for statistical significance for all analyses was set at a false discovery rate [33,34] of less than 0.20, calculated separately for overall cancer, cancer among non-hormone users, and ER+ breast cancers. In exploratory analyses, we also evaluated whether associations varied by time to diagnosis (<5 years vs ≥5 years), cancer stage (localized vs regional/distant), and menopausal hormone use (current vs non-current use) using the Wald test for heterogeneity. Interactions were declared if *p*-values were less than the false discovery rate threshold of 0.2.

To clarify whether metabolite-breast cancer associations were related to, or confounded by, established breast cancer risk factors, we also separately modeled associations when further adjusting for body mass index, smoking status, alcohol intake, history of diabetes, menopausal hormone therapy use, age at menarche, type of menopause, age at menopause, age at first live birth and number of births, history of benign breast disease, first-degree family history of breast cancer, and moderate-vigorous intensity physical activity (all modeled as indicated in Table 1). We then calculated the differences in ORs between the original and fully adjusted models as an indicator of the degree to which covariates are implicated in metabolite-breast cancer associations. Models adjusted for fewer covariates were also examined in supplementary analyses.

For the associations of hormonal metabolites with breast cancer risk, we additionally considered whether associations were independent of DHEA by adjusting for DHEA-sulfate, the body’s primary reservoir of DHEA. DHEA is the parent compound for many hormonal metabolites and is known to be associated with breast cancer risk [16,17,18].

Finally, to evaluate whether metabolite associations were independent of one another, we examined metabolite-metabolite correlations in a heatmap, ordered using hierarchical clustering. All statistical analyses were conducted in SAS 9.4 (SAS Institute, Cary, NC, USA) and RStudio version 1.1.453 (RStudio Inc., Boston, MA, USA).

## Figures and Tables

**Figure 1 metabolites-11-00095-f001:**
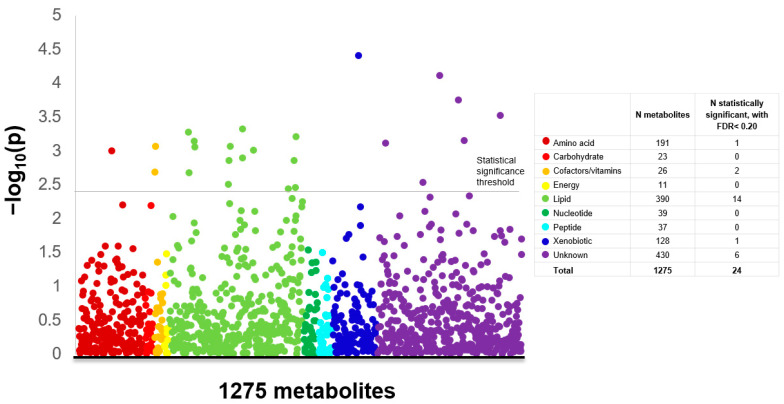
Manhattan plot displaying the *p*-values for metabolite-breast cancer risk associations from conditional logistic models, separated by chemical class. For each chemical class, the key shows the total number of metabolites and the number with statistically significant associations (false discovery rate <0.20). Models were adjusted for age at blood draw.

**Figure 2 metabolites-11-00095-f002:**
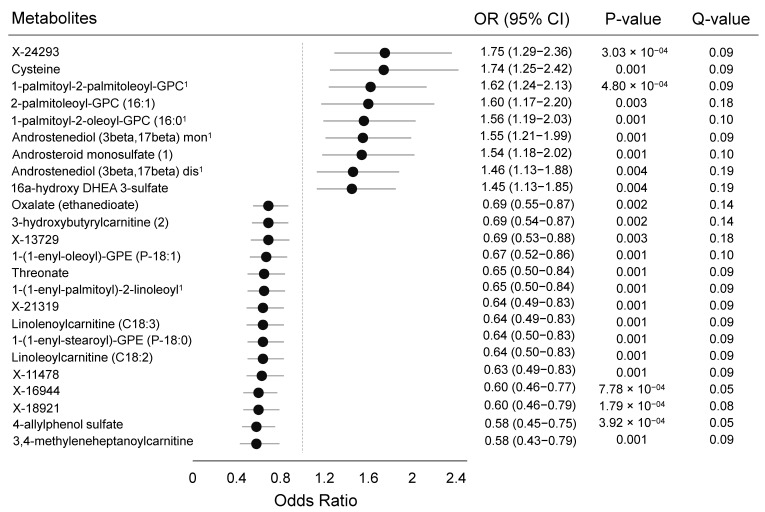
Odds ratios, 95% confidence intervals, *p*-values, and q-values for postmenopausal breast cancer when comparing the 90th with the 10th percentile levels of metabolites in conditional logistic models adjusted for age at blood draw. Results presented only for metabolites below the false discovery rate threshold of 0.20. Markers indicate the odds ratio, and error bars, the 95% confidence intervals. The q-value is the estimated probability of a false discovery. ^1^ Abbreviations in order of appearance: 1-palmitoyl-2-palmitoleoyl-GPC (16:0/16:1), 1-palmitoyl-2-oleoyl-GPC (16:0/18:1), Androstenediol (3beta,17beta) monosulfate (2), Androstenediol (3beta,17beta) disulfate (1), 1-(1-enyl-palmitoyl)-2-linoleoyl-GPC (P-16:0/18:2).

**Figure 3 metabolites-11-00095-f003:**
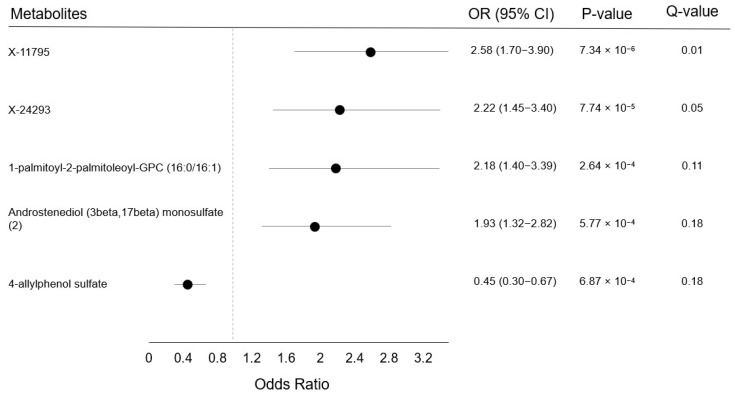
Odds ratios, 95% confidence intervals, *p*-values, and *Q*-values among non-hormone users at baseline for postmenopausal breast cancer when comparing the 90th with the 10th percentile levels of metabolites (n = 319 cases). Results were from unconditional logistic models adjusted for age at blood draw and are presented only for metabolites below the false discovery rate threshold of 0.20. Markers indicate the odds ratio, and error bars, the 95% confidence intervals. The *Q*-value is the estimated probability of a false discovery.

**Figure 4 metabolites-11-00095-f004:**
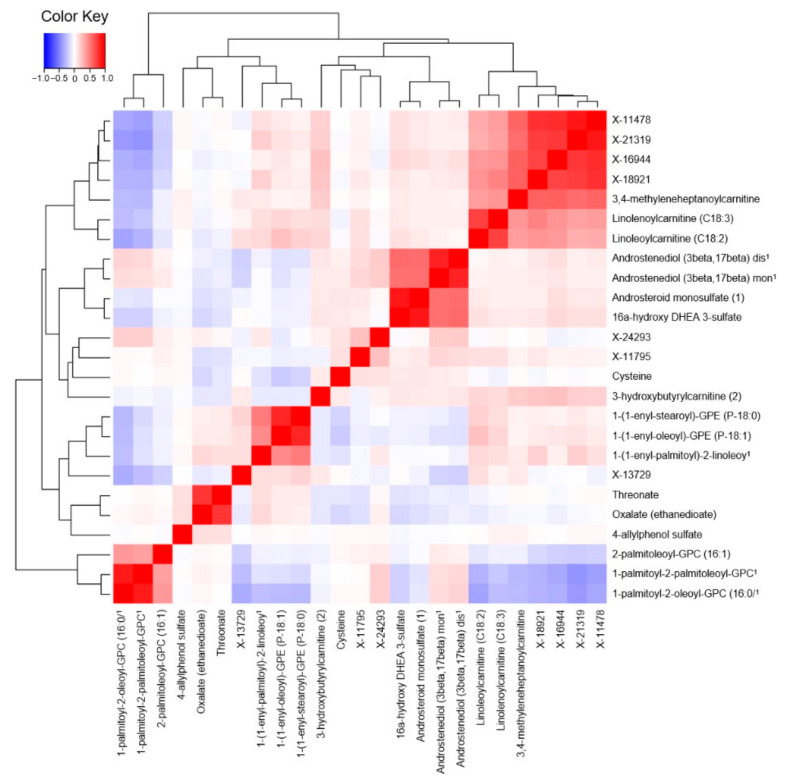
Heat map depicting pairwise correlations between the 24 metabolites associated with breast cancer risk. Metabolites were ordered using hierarchical clustering analysis as shown by light grey lines on the top and left sides. The key shows how colors correspond to the different Pearson correlation coefficients. For reference, the light red square for linoleoylcarnitine (C18:2) and 1-(1-enyl-oleoyl)-GPE (P-18:1) represents a correlation of 0.25, the medium red square for 16alpha-hydroxy DHEA 3-sulfate and androstenediol (3beta,17beta) monosulfate (2) represents a correlation of 0.55, and the dark red square for X-21319 and X-18921 represents a correlation of 0.75. The diagonal dark red line is the identify line, i.e., the correlation of 1 when comparing a metabolite against itself. ^1^ Abbreviations in order of appearance (left to right): 1-palmitoyl-2-oleoyl-GPC (16:0/18:1), 1-palmitoyl-2-palmitoleoyl-GPC (16:0/16:1), 1-(1-enyl-palmitoyl)-2-linoleoyl-GPC (P-16:0/18:2), Androstenediol (3beta,17beta) monosulfate (2), Androstenediol (3beta,17beta) disulfate (1).

**Table 1 metabolites-11-00095-t001:** Participant characteristics at blood draw in a nested case-control study in the Cancer Prevention Study-II *^,†^.

	Cases (n = 782)	Controls (n = 782)	*p* ^‡^
Age (mean yrs ± SD)	68 ± 6	68 ± 6	Matched
Body Mass Index (kg/m^2^)			0.21
<25	399 (51%)	426 (55%)	
25–29.9	246 (32%)	244 (31%)	
30+	132 (17%)	109 (14%)	
Race/ethnicity			Matched
Non-Hispanic white	764 (98%)	768 (98%)	
Other	18 (2%)	14 (2%)	
Smoking status			<0.0001
Never	384 (49%)	462 (59%)	
Former	363 (46%)	281 (36%)	
Current	25 (3%)	33 (4%)	
Alcohol intake			0.005
Non-drinker	254 (33%)	299 (38%)	
Current drinker	513 (66%)	449 (57%)	
History of diabetes			0.41
No	715 (91%)	722 (92%)	
Yes	42 (5%)	35 (5%)	
Menopausal hormone therapy use			0.06
Never	164 (21%)	192 (25%)	
Former	155 (20%)	170 (22%)	
Current	454 (58%)	406 (52%)	
Age at menarche (years)			0.61
≤11	29 (4%)	23 (3%)	
12–13	545 (70%)	542 (69%)	
14+	196 (25%)	207 (27%)	
Type of menopause, age at menopause (years)			0.04
Natural and <45	28 (4%)	40 (5%)	
Natural and 45–49	91 (12%)	111 (14%)	
Natural and 50–54	311 (40%)	268 (34%)	
Natural 55+	85 (11%)	73 (9%)	
Oophorectomy or surgery	226 (29%)	255 (33%)	
Drugs/treatment induced	0 (0%)	0 (0%)	
Age at first live birth (years), number of live births			0.03
Nulliparous	80 (10%)	63 (8%)	
<20 and 1+ children	37 (5%)	58 (7%)	
20–29 and 1–2 children	193 (25%)	183 (23%)	
20–29 and 3+ children	382 (49%)	409 (52%)	
30+ and 1+ children	86 (11%)	66 (8%)	
History of benign breast disease			0.30
No	528 (68%)	545 (70%)	
Yes	250 (32%)	230 (30%)	
Family history of breast cancer			0.11
No	554 (71%)	573 (73%)	
Yes	175 (22%)	148 (19%)	
Moderate-vigorous intensity physical activity (hours/week)			0.55
None/week	398 (51%)	386 (49%)	
<1 h/week	104 (13%)	111 (14%)	
1 h/week	78 (10%)	66 (8%)	
2–3 h/week	129 (17%)	150 (19%)	
4+ h/week	68 (9%)	66 (8%)	

* Due to the incidence density sampling, 17 participants are included in both case and control columns. ^†^ Percent estimates do not include missing data. The numbers of participants missing data were as follows: Body mass index: n = 8; Smoking status: n = 16; Alcohol intake: n = 49; History of diabetes; n = 50; Menopausal hormone therapy use: n = 50; Age at menarche: n = 22; Type of menopause and age at menopause: n = 76; Age at first live birth: n = 7; History of benign breast disease: n = 11; Family history of breast cancer: n = 114; Moderate-vigorous intensity physical activity: n = 8. ^‡^
*p* values derived from chi-square test.

**Table 2 metabolites-11-00095-t002:** Comparison of odds ratios (OR) and 95% confidence intervals (CIs) of breast cancer risk from age-adjusted versus multivariate models.

Metabolite	Age-Adjusted OR (95% CI)	Multivariate ^2^ OR (95% CI)	Difference in OR, %
X-24293	1.75 (1.29–2.36)	1.58 (1.13–2.22)	−9.7%
Cysteine	1.74 (1.25–2.42)	1.74 (1.21–2.52)	0.0%
1-palmitoyl-2-palmitoleoyl-GPC ^1^	1.62 (1.24–2.13)	1.58 (1.16–2.16)	−2.5%
2-palmitoleoyl-GPC (16:1)	1.60 (1.17–2.20)	1.52 (1.08–2.12)	−5.0%
1-palmitoyl-2-oleoyl-GPC (16:0) ^1^	1.56 (1.19–2.03)	1.52 (1.13–2.05)	−2.6%
Androstenediol (3beta,17beta) mon ^1^	1.55 (1.21–1.99)	1.45 (1.10–1.93)	−6.5%
Androsteroid monosulfate (1)	1.54 (1.18–2.02)	1.50 (1.11–2.03)	−2.6%
Androstenediol (3beta,17beta) dis ^1^	1.46 (1.13–1.88)	1.36 (1.02–1.81)	−6.9%
16alpha-hydroxy DHEA 3-sulfate	1.45 (1.13–1.85)	1.46 (1.11–1.93)	0.7%
Oxalate (ethanedioate)	0.69 (0.55–0.87)	0.66 (0.51–0.86)	−4.4%
3-hydroxybutyrylcarnitine (2)	0.69 (0.54–0.87)	0.73 (0.56–0.95)	5.8%
X-13729	0.69 (0.53–0.88)	0.72 (0.55–0.95)	4.4%
1-(1-enyl-oleoyl)-GPE (P-18:1)	0.67 (0.52–0.86)	0.69 (0.51–0.92)	3.0%
Threonate	0.65 (0.50–0.84)	0.61 (0.46–0.81)	−6.2%
1-(1-enyl-palmitoyl)-2-linoleoy ^1^	0.65 (0.50–0.84)	0.60 (0.45–0.81)	−7.7%
X-21319	0.64 (0.49–0.83)	0.64 (0.48–0.86)	0.0%
Linolenoylcarnitine (C18:3)	0.64 (0.49–0.83)	0.68 (0.51–0.91)	6.3%
1-(1-enyl-stearoyl)-GPE (P-18:0)	0.64 (0.50–0.83)	0.69 (0.51–0.92)	7.8%
Linoleoylcarnitine (C18:2)	0.64 (0.50–0.83)	0.67 (0.50–0.90)	4.7%
X-11478	0.63 (0.49–0.83)	0.61 (0.46–0.82)	−3.2%
X-16944	0.60 (0.46–0.77)	0.58 (0.44–0.76)	−3.3%
X-18921	0.60 (0.46–0.79)	0.58 (0.43–0.77)	−3.3%
4-allylphenol sulfate	0.58 (0.45–0.75)	0.60 (0.45–0.80)	3.5%
3,4-methyleneheptanoylcarnitine	0.58 (0.43–0.79)	0.58 (0.42–0.81)	0.0%

^1^: Abbreviations in order of appearance: 1-palmitoyl-2-palmitoleoyl-GPC (16:0/16:1), 1-palmitoyl-2-oleoyl-GPC (16:0/18:1), Androstenediol (3beta,17beta) monosulfate (2), Androstenediol (3beta,17beta) disulfate (1), 1-(1-enyl-palmitoyl)-2-linoleoyl-GPC (P-16:0/18:2). ^2^: Multivariate ORs were estimated with conditional logistic regression adjusted for age (years), body mass index (<25 kg/m^2^, 25.0–30 kg/m^2^, ≥30 kg/m^2^, unknown), smoking status (never, former, current), alcohol intake (non-drinker, current drinker, unknown), history of diabetes (no, yes, unknown), menopausal hormone therapy use (never, former, current, unknown), age at menarche (≤12 years, 12–13 years, ≥14 years, unknown), type of menopause and age at menopause (natural and <45 years, natural and 45–49 years, natural and 50–54 years, natural and ≥55 years, bilateral oophorectomy/surgery, drugs/radiation, hysterectomy, or unknown), age at first live birth and number of live births (nulliparous, age ≤19 years and ≥1 live births, age 20–29 years with 1 or 2 live births, age 20–29 with ≥3 live births, age 30+ with ≥1 live births, unknown), history of benign breast disease (no, yes, unknown), first degree family history of breast cancer (no, yes, unknown), and moderate-vigorous intensity physical activity (none, <1 h/week, 1 h/week, 2 h/week, 3 h/week, ≥4 h/week, unknown).

## Data Availability

Data described in the manuscript and analytic code are not available to protect participant confidentiality and in adherence with institutional policies.

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
