# Peer review of "A Metabolomics Analysis of Postmenopausal Breast Cancer Risk in the Cancer Prevention Study II"

_metabolites, 2021, doi:10.3390/metabo11020095_

Round 1

Reviewer 1 Report

I really liked the article, it was written logically, consistently, not overloaded with tables and figures. I have a question for the authors: is information available to you about the histological type of breast cancer (ductal, lobular) and receptor status (estrogen, progesterone, HER 2+)? It would be interesting to understand whether the same metabolites in different cases will be associated with a decrease / increase in the rate of breast cancer.

Author Response

REVIEWER 1: I really liked the article, it was written logically, consistently, not overloaded with tables and figures. I have a question for the authors: is information available to you about the histological type of breast cancer (ductal, lobular) and receptor status (estrogen, progesterone, HER 2+)? It would be interesting to understand whether the same metabolites in different cases will be associated with a decrease / increase in the rate of breast cancer.

RESPONSE: We did not have information available on histological type of cancer or HER2+ receptor status.  We did have information on estrogen receptor status and presented these results for estrogen receptor (ER)+ cancers (lines 112-115 and Supplementary Table 3) but not ER- cancers, since case numbers were too small.  In prior analyses, we and others have found that progesterone receptor status is highly correlated with estrogen receptor status.  Given this correlation, our sample size was too small to meaningfully distinguish ER vs. PR effects and so we decided to focus on just estrogen receptor status.

Reviewer 2 Report

This is a interesting and strong exploratory paper into metabolomics and breast cancer. I have the following 6 questions:

  1. Please explain why you matched for date of birth, date of blood draw and also adjusted for age at blood draw. Isn’t it over adjusting for age in addition to matching by the two other dates?
  2. Please explain why you matched on race/ethnicity?
  3. Please include more rationale behind adjusting for DHEAS in addition to the other androgen metabolites. Why does it matter that it is their precursor?
  4. Are your numbers substantial when stratifying by hormone use and adjusting for covariates? is your smaple larger than prvious studies once you have stratfied? Please include the Ns for these tables/figures on subsets of the full sample.
  5. What were the within and between batch CVs of your quality controls?
  6. Are your results consistent if you classify the metabolites as a doubling in metabolite level (log base 2) instead of comparing the 90th to 10th percentile?

Author Response

This is an interesting and strong exploratory paper into metabolomics and breast cancer. I have the following 6 questions:

Please explain why you matched for date of birth, date of blood draw and also adjusted for age at blood draw. Isn’t it over adjusting for age in addition to matching by the two other dates?

RESPONSE: Age is one of the most important risk factors for breast cancer and our matching involves some imprecision (plus or minus 6 months).  Adjusting for age accounts for any residual confounding that occurs due to the imprecision in matching.

Please explain why you matched on race/ethnicity?

RESPONSE: Race/ethnicity is associated with levels of some metabolites and, in some studies, with risk of breast cancer.  Matching helps to address potential confounding by this factor and could result in more statistically efficient analyses for race-specific analyses in the future, such as if we contribute our data to a large-scale meta-analysis.

Please include more rationale behind adjusting for DHEAS in addition to the other androgen metabolites. Why does it matter that it is their precursor?

RESPONSE: We added the following text to the discussion to help explicate the rationale for these analyses and to describe the importance of our results: "While DHEA has previously been linked with breast cancer risk DHEA [16-18], the associations we identified were statistically independent of DHEA levels, much like in one of our prior analyses [14].  This suggests that the pathways these metabolites are part of could constitute novel targets for prevention efforts and that knowing levels of these metabolites could at least nominally improve breast cancer risk prediction models."

Are your numbers substantial when stratifying by hormone use and adjusting for covariates? is your sample larger than previous studies once you have stratified? Please include the Ns for these tables/figures on subsets of the full sample.

RESPONSE: There were 319 cases among non-hormone users, which is a relatively modest sample for this subgroup.  We note this as a limitation in the discussion as follows: “The sample size among non-hormone-users was relatively small at 319 cases.”  We added the N for this analysis to Figure 3 and Supplementary Table 2.

What were the within and between batch CVs of your quality controls?

RESPONSE: We added this to the methods section (lines 317-318) as follows: "The median within-batch coefficient of variation was 0.13 and the median between-batch coefficient of variation was 0.18."

Are your results consistent if you classify the metabolites as a doubling in metabolite level (log base 2) instead of comparing the 90th to 10th percentile?

RESPONSE: Yes, using a log base 2 transformation has virtually no effect on the ordering of metabolites or on their p-values.  For instance, the top 10 metabolites remain exactly the same, and the p-values as shown remain the same, given the number of digits of rounding used in the manuscript.  Given how little results changed, we do not describe this in the manuscript, but could at the editors’ discretion.

In brief, the reason for the lack of change is that the 90th vs. 10th percentile coding is, algebraically, a linear transformation of the data in which we divide by a constant (the difference between the 90th and 10th percentile for each metabolite).  Linear transformations have no effect on p-values.  The transformation that is of primary relevance, then, is our use of natural log transformations prior to 90th vs. 10th percentile coding.  The natural log uses a base of ~2.72, which has a similar effect on the spread of the data as using log base 2, as requested by the reviewer.  Consequently, p-values are nearly identical across the board.  The odds ratio per 1-unit change does change, of course.  For example, for 4-allyphenol sulfate, the odds ratio going from the 10th to the 90th percentile is 0.58 while the odds ratio on the log 2 scale is 0.85.  That’s because women at the 90th percentile have levels of 4-allyphenol sulfate that are ~10 times those at the 10th percentile, and so a 1-unit change on this scale represents a much larger change than a 1-unit change on the log 2 scale.  We use the 90 vs. 10 scaling because it allows us to anchor our choice of units in real-world data whereas the meaning of a doubling in levels of metabolites can vary wildly from metabolite to metabolite.

Reviewer 3 Report

This is an important paper on serum metabolites, measured by the Metabolon kit, and breast cancer risk among postmenopausal women in a nested / prospective case-control study (782 cases/controls) from the USA. The authors identify 24 metabolites that are associated with breast cancer risk at a false discovery rate <0.20. In addition to plausible findings on steroid hormones (that the authors consider as a validation of findings from previous studies), several carnitines and glycerolipids were discovered as potential metabolic biomarkers of breast cancer risk. The paper is very well written and the team of authors is leading in in the field. Study protocol, metabolomics component, and data analyses all have a very high standard. The paper is of great interest to different communities.

I have the following minor points:

  1. Table 1: There is no p-value for race/ethnicity, as the variable was a matching factor. The same is true for age to my understanding, but a p values is shown. I suggest p values for both and a footnote that they were matching factors.
  2. The authors refer to breast feeding as an important protective factor in their introduction. However, there is no data on breast feeding in table 1 or the multivariable statistical models. Was there a reason not to consider breast feeding? Could the authors explain why breast feeding was not accounted for / or discuss this as a limitation?
  3. The authors write in their very nice introduction that there is some hope that metabolomics may help explain unexplained excess breast cancer risk. Now the strongest findings in Table 2 are at ORs of 1.75 and 0.58, after a liberal correction for multiple testing at a q-value <0.2 (which I find OK). The authors write that, except for plausible/expectable steroid hormone metabolites, they replicated hardly any of the findings from 5 previous similar studies. With the modest associations in the present study and in previous ones and the lack of replications, how realistic is it that metabolomics may help to explain excess breast cancer risk beyond known risk factors to a substantial degree? A comment on this in the discussion would be great.

Author Response

REVIEWER 3: This is an important paper on serum metabolites, measured by the Metabolon kit, and breast cancer risk among postmenopausal women in a nested / prospective case-control study (782 cases/controls) from the USA. The authors identify 24 metabolites that are associated with breast cancer risk at a false discovery rate <0.20. In addition to plausible findings on steroid hormones (that the authors consider as a validation of findings from previous studies), several carnitines and glycerolipids were discovered as potential metabolic biomarkers of breast cancer risk. The paper is very well written and the team of authors is leading in the field. Study protocol, metabolomics component, and data analyses all have a very high standard. The paper is of great interest to different communities.

I have the following minor points:

Table 1: There is no p-value for race/ethnicity, as the variable was a matching factor. The same is true for age to my understanding, but a p values is shown. I suggest p values for both and a footnote that they were matching factors.

RESPONSE: The reviewer is correct: age was a matching factor and we should therefore treat this like we did race/ethnicity in Table 1.  Traditionally, in epidemiology, no p-values are reported for matched factors in a Table 1 since the analysis is distorted by the matching itself.  Accordingly, we now indicate “Matched” for Age in Table 1.

The authors refer to breast feeding as an important protective factor in their introduction. However, there is no data on breast feeding in table 1 or the multivariable statistical models. Was there a reason not to consider breast feeding? Could the authors explain why breast feeding was not accounted for / or discuss this as a limitation?

RESPONSE: We did not have any data on the months of breast feeding.  Our adjustment for parity therefore serves as an admittedly crude proxy for this.  We have now included this in the limitations as follows (lines 251-253): "Additionally, though we adjusted for parity, we could not directly adjust for months of breast feeding, which is an important breast cancer risk factor [4]."

The authors write in their very nice introduction that there is some hope that metabolomics may help explain unexplained excess breast cancer risk. Now the strongest findings in Table 2 are at ORs of 1.75 and 0.58, after a liberal correction for multiple testing at a q-value <0.2 (which I find OK). The authors write that, except for plausible/expectable steroid hormone metabolites, they replicated hardly any of the findings from 5 previous similar studies. With the modest associations in the present study and in previous ones and the lack of replications, how realistic is it that metabolomics may help to explain excess breast cancer risk beyond known risk factors to a substantial degree? A comment on this in the discussion would be great.

RESPONSE: We feel it would be premature to comment on this in a scientific paper since metabolomics capabilities are still expanding rapidly, in at least two ways.  First, more and more metabolites are being assessed.  From our own internal data on 25 studies conducted since 2011, the number of metabolites has doubled every ~4.5 years.  Second, owing to improvements in sensitivity, platforms are detecting metabolites at lower and lower levels of blood concentrations.  These metabolites differ in kind from those of higher concentrations, as shown in https://www.ncbi.nlm.nih.gov/pmc/articles/PMC4123034/.  Consequently, the capabilities of metabolomics may be fundamentally different a decade from now and so it would be wise to avoid sweeping generalizations at this time.